# Dynamic Linkage between Bitcoin and Traditional Financial Assets: A Comparative Analysis of Different Time Frequencies

**DOI:** 10.3390/e24111565

**Published:** 2022-10-30

**Authors:** Panpan Wang, Xiaoxing Liu, Sixu Wu

**Affiliations:** 1School of Economics and Management, Southeast University, Nanjing 211189, China; 2School of Urban and Regional Science, East China Normal University, Shanghai 200241, China

**Keywords:** bitcoin, ADCC-GARCH, diversifier, hedge, safe haven

## Abstract

This study employs the ADCC-GARCH approach to investigate the dynamic correlation between bitcoin and 14 major financial assets in different time-frequency dimensions over the period 2013–2021, for which the risk diversification, hedging and safe-haven properties of bitcoin for those traditional assets are further examined. The results show that, first, bitcoin is positively linked to risk assets, including stock, bond and commodity, and negatively linked to the U.S. dollar, which is a safe-haven asset, so bitcoin is closer in nature to a risk asset than a safe-haven asset. Second, the high short-term volatility and speculative nature of the bitcoin market makes its long-term correlation with other assets stronger than the short-term. Third, the positive linkage between the prices of bitcoin and risk assets increases sharply under extreme shocks (e.g., the outbreak of COVID-19 in early 2020). Fourth, bitcoin can hedge against the U.S. dollar, and in the long term, bitcoin can hedge against the Chinese stock market and act as a safe haven for the U.S. stock market and crude oil. However, for most other traditional assets, bitcoin is only an effective diversifier.

## 1. Introduction

Digital cryptocurrencies have rapidly entered the public view in recent years, and their market trading scale continues to expand. As a representative species in the cryptocurrency market, bitcoin has exhibited dramatic volatility since its inception, and its price fluctuations have long been a concern for both academia and practitioners. Due to the soaring price of bitcoin in recent years, more investors around the world are entering the bitcoin market, expecting to make profits while lacking a deep understanding of the price formation mechanism of bitcoin and its asset properties, thus, facing huge investment risks. To establish an analytical framework about the price formation mechanism of bitcoin, it is first necessary to define whether bitcoin is a risk or a safe-haven asset. Some argue that because bitcoin is completely decentralized and not controlled by a traditional central bank and because bitcoin supply is limited by its own protocol design to a fixed total of 21 million coins, bitcoin has a similar anti-inflation value to gold and is a safe-haven asset. However, there are also arguments that the bitcoin market is highly speculative and that there is a clear positive correlation between the prices of bitcoin and various risk assets, thus, making it more of a risk asset in nature.

Is bitcoin a risk asset or a safe-haven asset? What are the linkages between bitcoin and major global assets? What are the dynamics of these linkages over time? This paper aims to explore the asset properties of bitcoin from the perspective of its linkage with traditional financial assets. We use the asymmetric dynamic conditional correlation (ADCC)-GARCH approach to examine the dynamic correlations between bitcoin and various traditional assets in different time frequency dimensions and further explore bitcoin’s diversification, hedging and safe-haven properties for each asset based on the dynamic correlations between bitcoin and these assets. Our analysis not only helps to further clarify the price formation mechanism of bitcoin and its role in portfolio management and helps investors to reasonably hold digital cryptocurrencies for investment, but also helps policymakers improve the dynamic monitoring and risk management of the cryptocurrency market represented by bitcoin.

The remainder of the paper is structured as follows. Section 2 reviews the relevant literature. Section 3 uses the ADCC-GARCH approach to quantitatively measure the dynamic correlation between bitcoin and various traditional financial assets in different time-frequency dimensions. Section 4 further identifies bitcoin’s risk diversification, hedging and safe-haven capabilities for each traditional asset. Section 5 concludes.

## 2. Literature Review

Bitcoin is a digital currency and payment system created by Satoshi Nakamoto [1]. As the first decentralized digital cryptocurrency, bitcoin’s price and popularity have risen rapidly since its introduction in 2009. With the growing popularity of bitcoin worldwide, the economic and financial properties of bitcoin have begun to attract the attention of researchers [2,3,4]. The relevant literature focuses on whether bitcoin is a currency or an asset, what kind of asset bitcoin is, and what kind of risk–return properties bitcoin has.

The earlier literature focused on whether bitcoin was a currency or an asset. Undeniably, there are some commonalities between bitcoin and currency, but from the perspective of monetary function, bitcoin can only be used as a medium of exchange and not as a unit of account or a storage of value [5]; therefore, bitcoin does not have a complete form of currency. Glaser et al. [6] focused on the liquidity of bitcoin when it functions as a medium of exchange, and argued that the convertibility between bitcoin and traditional currencies gives bitcoin liquidity, but the limited supply of bitcoin limits its liquidity. Böhme et al. [7] argued that the liquidity of bitcoin can be significantly weakened due to the frequent delays in bitcoin transactions. However, because of the anonymity of user identities, the bitcoin protocol does not restrict international transfer operations to countries that are on watch lists or embargoed, which provides bitcoin with higher flexibility and liquidity than deposit currencies in supporting international transfers [7]. In terms of the attitudes of bitcoin holders, since most bitcoin users view bitcoin as an investment tool rather than a transactional payment tool [6], the market value of bitcoin is much higher than the size of the economic transactions it facilitates [5], making bitcoin more of a speculative investment tool than a currency. Luther and Salter [8] examined bitcoin’s ability to replace traditional currencies based on the increase in bitcoin app downloads after the Cyprus bailout announcement and found that the rise in bitcoin app downloads was insignificant, suggesting that bitcoin is not replacing the currencies of those countries whose domestic banks are in trouble.

After determining that bitcoin is more of an asset, scholars began comparing bitcoin to traditional assets in an attempt to generalize which asset, or class of assets, bitcoin is more comparable. Bitcoin is often analogized to gold in the literature, and is even referred to as digital gold or the new gold [9]. The similarities between bitcoin and gold are that both have a much higher market value than their intrinsic value, and both derive their market value from scarcity of supply and high mining costs; both have no national attributes, and their supply is not controlled by the government; gold was used as a medium of exchange during the gold standard period but was abandoned later due to lack of liquidity, and bitcoin is likely to face similar problems in the future as the scale of bitcoin users expands. However, there are also differences between bitcoin and gold; for example, people use gold mainly because of its function as a store of value, while the instability of bitcoin prices makes it difficult for it to perform value storage effectively [10]. Klein et al. [11] compared the return volatility of bitcoin and gold and their respective correlations with other asset prices and found that while both prices respond asymmetrically to market shocks, their respective correlations with other asset prices differ significantly, especially during market downturns. Shahzad et al. [12] compared the safe-haven, hedging and diversification properties of bitcoin and gold for the G7 stock markets and found that gold outperforms bitcoin in terms of safe-haven and hedging effectiveness and can provide higher conditional diversification benefits for stock investments than bitcoin, while Thampanya et al. [13] found that neither bitcoin nor gold is a good hedge for the Thailand stock market. Furthermore, Whelan [14] drew an analogy between bitcoin and the U.S. dollar, arguing that both are used as a medium of exchange, but the difference is that the dollar is backed by the government, whereas bitcoin is a private currency issued by the private sector, and, thus, the supply and governance mechanisms for both assets are different. Dyhrberg [10] examined whether bitcoin is more comparative to gold or the U.S. dollar, and found that the behavioral characteristics of bitcoin prices have both dollar- and gold-like components because bitcoin’s property falls between a pure medium of exchange and a pure store of value. As such, bitcoin can be classified as an asset that is between the U.S. dollar and gold, and can be used as an effective tool for portfolio management.

More recent studies have begun to focus on the risk–return properties of bitcoin and its risk diversification, hedging and safe haven properties for traditional assets. Although bitcoin has a higher volatility than traditional assets [15], the inclusion of bitcoin in a portfolio may still improve the portfolio’s risk–return performance [16,17,18,19]. Eisl et al. [20] used a CVaR approach to find that although the inclusion of bitcoin raises the conditional VaR of the portfolio, this additional risk is fully compensated by high returns, which ultimately improves the risk–return ratio. Dyhrberg [21] used daily frequency data to test the hedging effect of bitcoin on U.K. equities and the USD exchange rate and found that bitcoin can be used as a hedge for the FTSE index as well as the USD/EUR and USD/GBP exchange rates over the 2010–2015 period. Yang et al. [22] used a time-frequency domain approach to find that bitcoin has the capability to hedge against the currency market in the long term. Using a daily frequency sample over 2011–2015, Bouri et al. [23] found that bitcoin prices are negatively correlated with the Nikkei 225, MSCI Pacific and commodity indices and, therefore, have the ability to hedge against price fluctuations in these assets. Chan et al. [24] found that bitcoin can be used as an effective hedge for the U.S., European, Canadian, Japanese and Chinese stock markets in the monthly frequency dimension over the period 2010–2017. Wang et al. [25] found that cryptocurrencies are a safe haven rather than a hedge for most international stock indices, and the safe-haven properties are more pronounced in developed markets or markets with larger market capitalizations and higher liquidity. Shahzad et al. [26] showed that the safe-haven role of bitcoin for Chinese, U.S. and other developed and developing stock markets is time-varying and varies across different stock markets. Urquhart and Zhang [27] used high-frequency data to examine bitcoin’s hedging and safe-haven capabilities for foreign exchange and found that bitcoin can be a hedge for the CHF, EUR and GBP, a diversifier for the AUD, CAD and JPY, and a safe haven for the CAD, CHF and GBP in times of extreme market turmoil. Wang et al. [28] examined the mean and volatility spillovers between bitcoin and six major financial assets in China and found that bitcoin can hedge China’s stock, bond, and monetary markets and can serve as a safe haven for China’s monetary market. Smales [29] argued that bitcoin’s high volatility, low liquidity and high transaction costs (in terms of time and fees) compared to other assets preclude it from being considered a safe haven until its market matures. Kwapień et al. [3] analyzed detrended cross-correlations between cryptocurrency markets and some traditional markets (including stock, commodity and forex markets) and found that the levels of cross-correlations become higher in turbulent periods.

After the outbreak of the COVID-19 pandemic, the price dynamics and portfolio performance of cryptocurrencies during the pandemic attracted widespread attention [30]. Wątorek et al. [4] pointed out that the COVID-19 pandemic has had a significant impact on the cryptocurrency market, transforming cryptocurrencies from a hedging instrument for investors fleeing traditional markets into a part of the global market which is closely linked to traditional financial instruments including currency, stock and commodity. Using a network connectedness model, Balcilar et al. [31] found increasing risk spillovers between cryptocurrencies and global emerging stock markets following the COVID-19 pandemic outbreak, and that cryptocurrencies cannot serve as a diversifier for emerging stock markets in both the short and long term. Caferra and Vidal-Tomas [32] used the wavelet coherence approach and Markov switching autoregressive model to find that cryptocurrencies have some hedging properties against stock markets in response to shocks from the COVID-19 pandemic. Using the COVID-19 outbreak as a quasi-experiment, Grobys [33] used a difference-in-differences approach to test bitcoin’s performance in hedging U.S. stock market risk and found that bitcoin performs poorly in hedging the tail risk of the U.S. market. Conlon and McGee [2] evaluated the safe-haven capability of bitcoin against traditional assets during the sharp decline in global financial markets following the outbreak of COVID-19, and found that bitcoin is not a safe haven for the S&P 500 and that including bitcoin in an equity portfolio at this time would substantially increase the portfolio’s downward risk exposure. Conlon et al. [34] further found that bitcoin is not a safe haven for most international stock markets except for China’s CSI 300 index. Wen et al. [35] used time-varying parameter vector autoregression to find that bitcoin is not a safe haven for crude oil and stocks during the COVID-19 pandemic. Dutta et al. [36] also found that bitcoin is only a diversifier rather than a safe haven for crude oil during the COVID-19 pandemic.

Although there has been a rich literature on the economic and financial properties of bitcoin, several gaps still remain. First, the sample of relevant studies does not cover a wide enough range of asset classes, and, therefore, they fail to provide a comprehensive dissection of the linkage between bitcoin and various major global assets and the diversification, hedging and safe-haven properties of bitcoin for these assets. Most of the existing studies focus on analyzing the linkage between bitcoin and a specific class of financial assets and bitcoin’s diversification, hedging and safe-haven properties for that class of assets, including stocks [24,25,26,31,32,33,34], commodities [36], currencies [22,27], etc. Although the sample selected by Wang et al. [28] covers multiple asset classes, these assets are all Chinese assets and are not globally representative. Second, most of the existing studies have examined the risk–return characteristics of bitcoin and its correlation with other assets in a frequency-specific sample, while few have conducted comparative analyses at different time frequencies. The high short-term volatility and speculative nature of the bitcoin market is likely to impair the short-term correlation between bitcoin and other assets, resulting in the linkage between bitcoin and other assets exhibiting very different characteristics in different time frequency dimensions, and may even lead to changes in bitcoin’s risk diversification, hedging and safe-haven properties in different time frequency dimensions.

We selected the prices of bitcoin and 14 major financial assets covering stock, bond, commodity and currency over the period 2013–2021 as a sample to test the dynamic linkage of bitcoin with each asset and the linkage’s variation in different time frequency dimensions, and to further identify the risk diversification, hedging and safe-haven properties of bitcoin for various assets. We first used the ADCC-GARCH approach to quantitatively measure the dynamic correlation between bitcoin and other traditional assets. Based on the estimated dynamic conditional correlation (DCC) series, we further adopted Ratner and Chiu’s [37] approach to identify the risk diversification, hedging and safe-haven properties of bitcoin for various types of assets to assess the extent to which bitcoin can be used as a diversifier, hedge or safe haven for those assets. As a result, we show that: (i) bitcoin is positively linked to risk assets, including stocks, bonds and commodities, and negatively linked to the U.S. dollar, which is a typical safe-haven asset, so bitcoin is closer in nature to a risk asset than a safe-haven asset; (ii) the high short-term volatility and speculative nature of the bitcoin market makes its long-term correlation with other asset prices stronger than the short-term correlation; (iii) the positive linkage between the prices of bitcoin and risk assets increases sharply under extreme shocks (e.g., the outbreak of COVID-19 in early 2020); (iv) bitcoin can hedge against the U.S. dollar, and in the long term, bitcoin can hedge against the Chinese stock market and act as a safe haven for the U.S. stock market and crude oil. However, for most other traditional assets, bitcoin is only an effective diversifier.

The contribution of this study is twofold. First, our selected sample covers four major asset classes—stock, bond, commodity and currency—which specifically include 14 major representative global financial assets. Based on this sample, we provide a comprehensive analysis of the linkage between bitcoin and major global assets as well as bitcoin’s diversification, hedging and safe haven properties for these assets. Second, we also perform a comparative analysis in different time-frequency dimensions, comparing the dynamic correlation between bitcoin and each asset in daily, weekly and semi-monthly frequency dimensions as well as the differences in bitcoin’s risk diversification, hedging and safe-haven capabilities for other assets in different time-frequency dimensions.

## 3. The Dynamic Correlation between Bitcoin and Traditional Assets

### 3.1. ADCC-GARCH Model

The aim of this study is to examine the dynamic linkages between bitcoin and various traditional financial assets at different frequencies and to explore the risk diversification, hedging and safe-haven properties of bitcoin for each asset based on the dynamic linkages between bitcoin and them. To capture the time-varying correlation between bitcoin and other traditional financial assets, we employed the DCC-GARCH approach. The DCC-GARCH method can estimate the time-varying conditional correlation coefficient and has the advantage of portraying the dynamic relationship between variables. The DCC model was first introduced by Engle [38] to allow for time-varying correlation between variables. Cappiello et al. [39] further introduced an asymmetric version of the DCC-GARCH (i.e., ADCC-GARCH) to address the effect of asymmetric information on time-varying correlations. In this study, the ADCC model of Cappiello et al. [39] was used to model the volatility dynamics and conditional correlation between bitcoin and other assets.

Let $r_{t}$ be a n × 1 vector of asset returns. The AR(1) process for $r_{t}$ conditioned on the information set $I_{t-1}$ can be written as follows:(1)$$r_{t}=\mu +\varphi r_{t-1}+\varepsilon_{t}$$

The residuals are modeled as:(2)$$\varepsilon_{t}=H_{t}^{1/2}z_{t}$$

$H_{t}$ is the conditional covariance matrix of $r_{t}$, and $z_{t}$ is a n × 1 i.i.d. random vector of errors. Engle’s [38] DCC model is estimated in two steps, with the GARCH parameters estimated in the first step and the conditional correlation in the second step, where:(3)$$H_{t}=D_{t}R_{t}D_{t}$$
where $H_{t}$ is a n×n conditional covariance matrix, $R_{t}$ is the conditional correlation matrix, and $D_{t}$ is the diagonal matrix with time-varying standard deviations on the diagonal.
(4)$$D_{t}=diag\left(h_{1,t}^{1/2},\ldots ,h_{n,t}^{1/2}\right)$$
(5)$$R_{t}=diag\left(q_{1,t}^{-1/2},\ldots ,q_{n,t}^{-1/2}\right)Q_{t}diag\left(q_{1,t}^{-1/2},\ldots ,q_{n,t}^{-1/2}\right)$$

The expression for h is a univariate GARCH. For the GARCH(1,1) model, the elements of $H_{t}$ can be written as follows:(6)$$h_{i,t}=\omega_{i}+\alpha_{i}\varepsilon_{i,t-1}^{2}+\beta_{i}h_{i,t-1}$$

$Q_{t}$ is a symmetric positive definite matrix that can be written in the following form:(7)$$Q_{t}=\left(1-\theta_{1}-\theta_{2}\right)\bar{Q}+\theta_{1}z_{t}z_{t-1}^{\prime }+\theta_{2}Q_{t-1}$$
where $\bar{Q}$ is the n×n unconditional correlation matrix of the standardized residuals $z_{i,t}$ ($z_{i,t}=\varepsilon_{i,t}/\sqrt{h_{i,t}}$). The parameters $\theta_{1}$ and $\theta_{2}$ are non-negative and are related to the exponential smoothing process used to construct the dynamic conditional correlations. The DCC model is mean-reverting as long as $\theta_{1}$ + $\theta_{2}$ < 1. The correlation is estimated as:(8)$$\rho_{i,j,t}=\frac{q_{i,j,t}}{\sqrt{q_{i,i,t}q_{j,j,t}}}$$

Since the above DCC model does not allow for asymmetries and asset-specific news impact parameter, Cappiello et al. [39] developed the ADCC model to incorporate asymmetric effects and asset-specific news impact. For the ADDC model, the dynamics of Q is of the following form:(9)$$Q_{t}=\left(\bar{Q}-A^{\prime }\bar{Q}A-B^{\prime }\bar{Q}B-G^{\prime }{\bar{Q}}^{-}G\right)+A^{\prime }z_{t-1}z_{t-1}^{\prime }A+B^{\prime }Q_{t-1}B+G^{\prime }z_{t}^{-}z_{t}^{\prime -}G$$
where A, B and G are n×n parameter matrices and $z_{t}^{-}$ is the zero-threshold standardized error, which is equal to $z_{t}$ when less than zero, and zero otherwise. $\bar{Q}$ and ${\bar{Q}}^{-}$ are the unconditional matrices of $z_{t}$ and $z_{t}^{-}$, respectively.

### 3.2. Data and Descriptive Statistics

We selected daily data of bitcoin and 14 asset prices from 6 June 2013, to 2 August 2021 as the sample, with the number of observations being 1703. Data were obtained from the Wind and Yahoo Finance databases. These assets cover four categories, namely, stock, bond, commodity and currency. The stock sample included the MSCI world index, S&P 500, FTSE 100, DAX 30, Nikkei 225 and SSEC. The bond sample includes the US bond index, non-U.S. bond index and emerging markets bond index, which are measured by the prices of ETFs that track the three indices. The commodity sample includes the S&P GSCI, CRB commodity index, Brent oil and gold. The currency sample is the U.S. dollar index.

Figure 1 plots the time series of the prices of bitcoin versus each asset. The correlation between bitcoin and most asset prices is not stable, with some periods moving in the same direction and some moving in the opposite direction. For example, bitcoin was positively correlated with the S&P 500 during 2017–2018 and then became negatively correlated in 2019. This suggests that the linkages between bitcoin and these traditional asset prices are time-varying, thus, necessitating the use of an (asymmetric) DCC model to more accurately capture the dynamic correlation between bitcoin and each asset. The daily percentage returns of all asset prices are calculated based on the following equation:(10)$$r_{t}=100*\left(\ln P_{t}-\ln P_{t-1}\right)$$
where $r_{t}$ represents the return of each asset and $P_{t}$ represents the original asset price. Table 1 presents descriptive statistics for the return of each asset. Bitcoin had a much higher mean return than other assets, which reflected the long-term upward trend in bitcoin prices over the sample period. Bitcoin also had a much higher standard deviation than other assets, indicating that bitcoin prices were extremely volatile. The level of skewness in all asset returns was not high, except for the emerging markets bond index, which exhibited a clear left skew. Most asset returns had high kurtosis. The Jarque–Bera (J-B) test significantly rejected the assumption of normality of the distribution for all series.

GARCH modeling requires data stationarity to ensure the validity of the estimation, and having ARCH effects is also a prerequisite for GARCH modeling. We used the augmented Dickey–Fuller (ADF) and Phillips–Perron (PP) tests to perform unit root tests on the return series (i.e., the first-order difference of asset price) and the Portmanteau test to perform an ARCH effect test on the return series, with the results reported in Table 2 and Table 3, respectively. All return series were stationary at the 1% significance level and had significant ARCH effects at lags of order 5, 10 and 15, thus, satisfying the GARCH modeling conditions.

Given the extremely high short-term volatility of bitcoin prices, bitcoin’s short-term correlation with other assets is likely to be disturbed by its sharp short-term volatility. Intuitively, the long-term correlation between bitcoin and other asset prices is likely to be more stable than the short-term correlation. To this end, we not only performed ADCC-GARCH analysis on daily frequency data of bitcoin and other asset prices, but also further performed ADCC-GARCH analysis on weekly and semi-monthly frequency data and then compared the results at different time frequencies to examine the differences in the linkage between bitcoin and various assets at different time frequencies. The weekly and semi-monthly frequency samples were obtained by taking the weekly and semi-monthly end-of-period values of the daily frequency samples, respectively, and the number of observations for both samples was 422 and 196, respectively. In addition to the weekly and semi-monthly frequencies, we also established an ADCC-GARCH model for the monthly frequency sample. However, since the number of observations for the monthly sample was only 97, the algorithm could not converge when performing ADCC-GARCH estimation. Therefore, it was abandoned.

### 3.3. ADCC-GARCH Estimation Results

The mean equation of the GARCH model was set to be AR(1) with the intercept term included, and the parameters of the intercept term and AR(1) were denoted by μ and φ, respectively. The variance equation was set to be GARCH(1,1), and the parameters of its intercept term, ARCH term and GARCH term were denoted by ω, α and β, respectively. The dynamic correlation parameters of the ADCC model were denoted by a, b and g, and υ denoted the joint distribution parameter of the model. Since all return series were not normally distributed, the multivariate joint t-distribution was selected for the distribution function.

Table 4 reports the ADCC-GARCH estimation results for the daily frequency sample of bitcoin and each asset price. In the variance equation, the coefficients of the ARCH and GARCH terms for all assets were significantly positive at least at the 5% level, indicating that the GARCH(1,1) setting was plausible. The coefficients of the GARCH term for all assets were much larger than the coefficients of the ARCH term, indicating that the conditional variance was more influenced by its prior period value and less sensitive to the previous period’s return volatility, which showed that the price movements of these assets exhibited volatility clustering. From the ADCC estimation results, a was not negative, indicating that the standardized residuals with one lag had a positive effect on the dynamic correlation coefficient; b was close to 1, indicating that the dynamic correlation between bitcoin and other assets had strong persistence; and the sum of a and b was less than 1, ensuring that the conditional covariance matrix was positive definite and mean-reverting. ARCH tests were further performed on the residual terms of the ADCC-GARCH estimation results, and the results showed no significant ARCH effect. The above results indicated that the ADCC-GARCH estimation results were reliable. In addition to the daily frequency sample, we also performed ADCC-GARCH modeling for the weekly and semi-monthly frequency samples.

Figure 2 shows the trend of dynamic correlation coefficients between bitcoin and various assets, including daily, weekly and semi-monthly frequencies. The correlation coefficients between bitcoin and each asset all exhibited significant time variability, suggesting that using an ADCC-GARCH approach was necessary to capture the dynamic correlation between bitcoin and each asset. We classified these assets into stock, bond, commodity and currency to further analyze the dynamic correlation between bitcoin and different classes of assets.

Figure 2, panels (1)–(6), show the trend of dynamic correlation coefficients between bitcoin and representative stock indices. Firstly, in the daily frequency dimension, the correlation coefficients between bitcoin and all stock indices were low and largely fluctuated around 0, showing no sustained positive or negative correlation. The reason for this may be that bitcoin’s high short-term volatility undermines its short-term correlation with other assets. Secondly, as the frequency changed from high to low, bitcoin began to show a significant positive correlation with most stock indices. In both the weekly and semi-monthly frequency dimensions, the correlation coefficients between bitcoin and global, U.S., U.K., German and Japanese stock indices were consistently positive in most periods, and the magnitude of the coefficients was also significantly higher. In particular, the dynamic correlation coefficients of bitcoin with global, U.S. and U.K. stock indices showed a clear “semimonthly frequency > weekly frequency > daily frequency”. This showed that bitcoin had a weak correlation with major stock prices in the short term, but a more stable positive correlation in the long term. Thirdly, bitcoin’s linkage with SSEC had different characteristics from its linkage with stock indices of developed countries. While the correlation between bitcoin and SSEC in the daily frequency dimension fluctuated around 0 over the full sample interval, the daily frequency correlation between the two was negative for most periods before 2017, which was consistent with the findings of Wang et al. [28]. Furthermore, the dynamic correlation coefficient between bitcoin and SSEC in the semi-monthly frequency dimension was negative in most periods, indicating that bitcoin and SSEC are negatively correlated in the long term. Unlike developed countries, China’s financial markets have long been subject to capital controls, resulting in relatively limited channels for investors to invest abroad. In this context, when there is a long-term downward trend in the Chinese stock market, investors tend to enter the cryptocurrency (e.g., bitcoin) market to hedge their domestic stock investment losses, which is a possible reason for the negative correlation between bitcoin and the SSEC index in the long run. Fourth, the bitcoin-stock linkage increased sharply when subjected to exogenous extreme shocks. This observation was consistent with Kwapień et al. [3], who also found that the level of correlation between the cryptocurrency market and the stock market becomes higher during turbulent periods. Following the outbreak of COVID-19 in early 2020, the dynamic correlation coefficients between bitcoin and all stock indices rose rapidly, with the weekly frequency correlation coefficients of bitcoin with global, U.S., Japanese and Chinese stock indices rising sharply to approximately 0.5, 0.35, 0.7 and 0.25, respectively, and the semi-monthly frequency correlation coefficients with the U.K. and German stock indices both rising sharply to levels close to 0.8. The epidemic shock has led to a rapid rise in uncertainty and a sudden drop in investor risk appetite, causing investors to be less willing to hold not only traditional risk assets such as stocks, but also bitcoin, which has led to a sharp decline in both bitcoin and stock markets and a sharp increase in the positive linkage between bitcoin and the underlying stock indices.

Figure 2, panels (7)–(9), show the trend of dynamic correlation coefficients between bitcoin and representative bond indices. Firstly, similar to the dynamic correlation coefficient between bitcoin and stock prices, the dynamic correlation coefficient between bitcoin and bond prices exhibited a gradual increase from the short term (high frequency) to the long term (low frequency). In the daily frequency dimension, the dynamic correlation coefficients between bitcoin and the U.S. bond index, the non-U.S. bond index and the emerging markets bond index all fluctuated in a small range around 0; however, in the weekly and semi-monthly frequency dimensions, the coefficients were not only consistently positive but also significantly higher in magnitude. In particular, the dynamic correlation coefficients between bitcoin and both the non-U.S. bond index and emerging markets bond index showed a clear “semimonthly frequency > weekly frequency > daily frequency”. Secondly, in the comparable time-frequency dimension, the correlation between bitcoin and bond prices was lower than its correlation with stock prices, indicating that bitcoin is more closely linked to the stock market than its linkage to the bond market. Thirdly, the linkage between bitcoin and bond prices also exhibited a sharp enhancement in response to extreme shocks. The outbreak of COVID-19 in early 2020 caused both bitcoin and bond prices to fall significantly, resulting in the weekly frequency correlation coefficients of bitcoin with the U.S. bond index, the daily frequency correlation coefficients with the non-U.S. bond index and the semi-monthly frequency correlation coefficients with the emerging markets bond index rising sharply to over 0.5, 0.3 and 0.6, respectively.

Figure 2, panels (10)–(13), show the trend of dynamic correlation coefficients between bitcoin and representative commodity prices. Firstly, with the exception of gold, the dynamic correlation coefficients between bitcoin prices and the S&P GSCI, CRB commodity index and oil prices were positive for all periods and all time-frequency dimensions, indicating a persistent positive linkage between bitcoin and major commodity prices. Secondly, similar to the dynamic correlation coefficient between bitcoin and stocks/bonds, the dynamic correlation coefficients between bitcoin and the three commodities other than gold showed a gradual increase from the short term (high frequency) to the long term (low frequency). In the daily frequency dimension, the dynamic correlation coefficients between bitcoin and the three commodities, although consistently positive, were at a low level of below 0.1 for most periods, while in the weekly and semi-monthly frequency dimensions, the positive correlation coefficients were significantly higher. The correlation between bitcoin and gold on both daily and weekly frequencies fluctuated basically in a small range around 0. However, the correlation between the two on semi-monthly frequencies, increased. Thirdly, the bitcoin–commodity market linkage also exhibited a sharp increase in response to extreme shocks, which was consistent with Kwapień et al. [3], who showed that the level of correlation between the cryptocurrency market and the commodity market becomes higher during turbulent periods. Following the outbreak of COVID-19 in early 2020, both bitcoin and commodity prices fell rapidly, with bitcoin’s semi-monthly frequency correlation coefficients with the S&P GSCI, oil prices and gold prices rising sharply to over 0.6, 0.5 and 0.6, respectively, and its weekly frequency correlation coefficient with the CRB commodity index rising rapidly to approximately 0.35.

Figure 2, panel (14), shows the trend of the dynamic correlation coefficient between bitcoin and the U.S. dollar index. At all frequencies, the dynamic correlation coefficient between bitcoin and the U.S. dollar index was negative in most periods, indicating that bitcoin price has an inverse linkage to the U.S. dollar index and that bitcoin can be used as an effective hedge against dollar depreciation. Bitcoin had a weak negative correlation with the U.S. dollar index on the daily and weekly frequencies, but showed a strong negative correlation for most periods on the semi-monthly frequency, peaking at nearly −0.4. The reason for the negative correlation between bitcoin and the dollar index may be interpreted in two ways. First, since bitcoin is denominated in USD, a dollar depreciation will cause bitcoin to become cheaper, thus, increasing demand for bitcoin and driving its price upward. Second, as deduced from the previous results regarding the predominantly positive correlation between bitcoin and stock/bond/commodity prices, bitcoin is closer in nature to a risk asset, while the U.S. dollar is typically a safe-haven asset, so the two prices naturally exhibit a negative correlation.

In summary, the linkage between bitcoin and various assets varies by asset class and time frequency, and can undergo significant structural changes in response to exogenous shocks in international financial markets. In terms of asset classes, bitcoin was positively correlated with risk assets including stock, commodity and bond, and bitcoin’s positive correlation with stock or commodity was stronger than its positive correlation with bond; bitcoin had a significant negative correlation with the U.S. dollar, a typical safe-haven asset. As such, bitcoin is closer in nature to a risk asset than a safe-haven asset. In terms of time frequency, the long-term correlation between bitcoin and various asset prices was significantly stronger than the short-term correlation, mainly because the short-term high volatility and speculative nature of the bitcoin market undermine its short-term correlation with other assets. Finally, the linkage between bitcoin and the risk assets can increase sharply in response to exogenous extreme shocks. For example, after the outbreak of COVID-19 in early 2020, the plunge in investor risk appetite led to a sharp decline in the prices of bitcoin and risk assets (including stocks, bonds and commodities), at which point the positive linkage between bitcoin and those risk assets rose rapidly. This also suggests that the outbreak of the COVID-19 pandemic has accelerated the integration of bitcoin with traditional financial markets, transforming it into part of a global market that is increasingly correlated with traditional assets [4].

## 4. Bitcoin’s Risk Diversification, Hedging and Safe-Haven Properties for Other Assets

The DCC coefficients estimated in the previous section can characterize the dynamic linkage between bitcoin and other assets and provide a basis for further identifying the risk diversification, hedging and safe-haven properties of bitcoin for those assets.

### 4.1. Criteria for Distinguishing an Asset’s Risk Diversification, Hedging and Safe-Haven Properties

We first established the criteria by which bitcoin is considered a diversifier, hedge or a safe haven. We followed Baur and Lucey [40] and Ratner and Chiu [37] to define a diversifier, hedge and safe haven, as these have become standard in the literature. Following Baur and Lucey [40] and Ratner and Chiu [37], we then distinguished bitcoin’s risk diversification, hedging and safe-haven properties. The risk diversification property means that bitcoin is positively (but not perfectly) correlated with the returns of other assets. The hedging property includes a weak hedge, which means that bitcoin is uncorrelated with the returns of other assets, and a strong hedge, which means that bitcoin is negatively correlated with the returns of other assets. The safe-haven property includes a weak safe haven, which means that bitcoin is uncorrelated with the returns of other assets in times of market turmoil, and a strong safe haven, which means that bitcoin is negatively correlated with the returns of other assets in times of market turmoil. It is worth noting that our definition of a safe haven is ex post facto; that is, an asset (such as bitcoin) is considered a safe haven if investors show a preference for it during a market stress, which is of course only a necessary but not sufficient condition for an asset to be a safe-haven asset.

### 4.2. Identification Approach

The DCC coefficient between bitcoin and each asset is regressed on the lower quantile dummy variables of the corresponding asset returns to specifically identify the diversification, hedging and safe-haven properties of bitcoin for that asset. The regression equation is as follows:(11)$$DCC_{t}=m_{0}+m_{1}D\left(r_{other\;asset}q_{10}\right)+m_{2}D\left(r_{other\;asset}q_{5}\right)+m_{3}D\left(r_{other\;asset}q_{1}\right)+\varepsilon_{t}$$
where $r_{other\;asset}$ is the return on the corresponding asset, and $D\left(r_{other\;asset}q_{10}\right)$, $D\left(r_{other\;asset}q_{5}\right)$ and $D\left(r_{other\;asset}q_{1}\right)$ denote the 10%, 5%, and 1% lower quartile dummy variables for the return on that asset, respectively. Figure 3 shows the 10%, 5% and 1% lower quartile values of daily returns for 14 traditional financial assets. The observations below the lower quartile values comprise the analysis period for our test of bitcoin’s safe-haven properties. For observations below the 10% lower quantile value, we set $D\left(r_{other\;asset}q_{10}\right)$ = 1; for observations below the 5% lower quantile value, we set $D\left(r_{other\;asset}q_{5}\right)$ = 1; and for observations below the 1% lower quantile value, we set $D\left(r_{other\;asset}q_{1}\right)$ = 1.

If $m_{0}$ is significantly positive, then bitcoin has risk diversification capability for that asset; if $m_{0}$ is equal to 0, then bitcoin has weak hedging capability for that asset; if $m_{0}$ is negative, then bitcoin has strong hedging capability for that asset. If $m_{1}$, $m_{2}$ and $m_{3}$ are not significantly nonzero, then bitcoin has a weak safe-haven capability for that asset; if $m_{1}$, $m_{2}$ and $m_{3}$ are significantly negative, then bitcoin has a strong safe-haven capability for that asset.

### 4.3. Regression Results

We extracted the pairwise DCC series between bitcoin and other assets from the ADCC-GARCH estimation results and then regressed the DCC series based on Equation (11) to assess the diversification, hedging and safe-haven properties of bitcoin for each asset. For example, by regressing the DCC series between bitcoin and the MSCI world index on a constant ($m_{0}$) and three dummy variables ($m_{1}$, $m_{2}$ and $m_{3}$), we can assess the diversification or hedging capability of bitcoin for the global stock market based on the value and significance level of $m_{0}$ and the safe-haven capability of bitcoin for the global stock market based on the values and significance levels of $m_{1}$, $m_{2}$ and $m_{3}$. Table 5, Table 6 and Table 7 report the regression results for the daily, weekly and semi-monthly DCC series, respectively.

The regression results for the daily frequency sample (Table 5) show that bitcoin cannot be considered a strong safe haven for all financial assets, implying that investors holding bitcoin cannot protect against extreme volatility in the prices of those assets. For the S&P 500, FTSE 100, DAX30, U.S. bond index, non-U.S. bond index, emerging markets bond index, S&P GSCI, oil and gold, the coefficients of the dummy variables characterizing the lower quartile values of their returns ($m_{1}$, $m_{2}$ and $m_{3}$) are all significantly positive in at least one case, which can only mean that bitcoin is an effective risk diversifier at the corresponding lower quartile levels of these asset returns. In terms of hedging properties, only the constant term ($m_{0}$) of the U.S. dollar index is significantly negative across all assets, implying that bitcoin is an effective tool to hedge against the movement of the USD exchange rate and that USD investors in the FX market tend to use bitcoin to hedge their currency portfolios. This is consistent with the findings of Dyhrberg [21], who also shows that bitcoin can be used as a hedge for the USD/EUR and USD/GBP exchange rates. All assets other than the U.S. dollar have significantly positive $m_{0}$, so for these assets, bitcoin is only an effective risk diversifier. The main reason why bitcoin has risk diversification capabilities for most assets is that the mechanism of bitcoin price formation is more influenced by the bitcoin market’s own supply and demand and investors’ preferences for cryptocurrencies and less correlated with global macroeconomic and financial developments.

The regression results for the weekly frequency sample (Table 6) showed that in terms of hedging properties, only the U.S. dollar index had a significantly negative $m_{0}$, while all other assets had a significantly positive $m_{0}$. This suggests that in the weekly frequency dimension, bitcoin is still only a strong hedge against the U.S. dollar exchange rate and only an effective risk diversifier for other assets. In terms of safe-haven properties, $m_{3}$ was significantly negative for the S&P 500, and $m_{1}$ was significantly negative for crude oil, suggesting that bitcoin can be viewed to some extent as a safe haven against extreme volatility in the U.S. stock market (at the lower 1% quartile) and crude oil market (at the lower 10% quartile). This also means that investors have a tendency to put their money into the bitcoin market when there is a crisis in the U.S. stock market and the crude oil market. A possible explanation for why investors choose bitcoin as a safe haven in some cases, is that bitcoin operates as a decentralized cryptocurrency that is completely independent of any central institution. When traditional financial markets are under downward pressure, investors choose to seek shelter in the bitcoin market, which is independent of the traditional financial system and its underlying technical architecture.

The regression results for the semi-monthly frequency sample (Table 7) were similar to those for the weekly frequency samples. However, there were two differences. First, in terms of hedging properties, on the semi-monthly frequency, bitcoin not only had the ability to hedge the U.S. dollar index but also had a significant hedging effect on the SSEC. This suggests that bitcoin has the ability to hedge against declines in China’s stock market in the long term. Moreover, the observation that bitcoin can hedge the Chinese stock market in the long term is also mentioned in the literature; for example, Chan et al. [24] found that bitcoin could serve as an effective hedge for the Chinese stock market in the monthly frequency dimension over the period 2010–2017. The likely reason is, that due to the existence of cross-border capital flow controls in China, when its domestic stock market is under long-term downward pressure, investors may seek to enter the bitcoin market to hedge against declines in the domestic stock market, as they have limited access to foreign investment. Second, in terms of safe-haven properties, on the semi-monthly frequency, bitcoin no longer has a strong hedging effect on the crude oil market but can still be considered to some extent as a safe haven against the risk of extreme volatility in the U.S. stock market at the low 1% quartile. Combining the weekly and semi-monthly frequency regression results, it was found that among the global stock markets, bitcoin had a significant safe-haven effect only for the U.S. stock market, which echoed the results found by Wang et al. [25] that the safe-haven properties of cryptocurrencies are more pronounced in developed markets or markets with larger market capitalization and higher liquidity.

In summary, bitcoin’s risk diversification, hedging and safe-haven properties vary across time frequency dimensions, so distinguishing the holding period matters to bitcoin holders. For example, bitcoin does not have hedging ability against the Chinese stock market on the daily and weekly frequencies, but starts to produce a significant hedging effect on the semi-monthly frequency. In addition, bitcoin has no safe-haven properties for all assets on the daily frequency, but is starting to exhibit safe-haven properties for some assets (i.e., S&P 500 and crude oil) on the weekly and semi-monthly frequencies. One explanation for the differences in bitcoin’s hedging and safe-haven properties at different time frequencies is that bitcoin’s high short-term volatility and its strong speculative properties undermine its hedging and safe-haven properties in the short term and may even compromise its hedging and safe-haven properties in the long term. Another explanation is that bitcoin’s hedging and safe-haven properties at different frequencies may be driven by different factors, as price formation of bitcoin may be influenced by different factors in the long and short term.

## 5. Conclusions

Using bitcoin and 14 global financial asset price data covering stock, bond, commercial and currency for the period 2013–2021, this study applied the ADCC-GARCH approach to test the dynamic correlation between bitcoin and each asset at different time frequencies, and further identified the risk diversification, hedging and safe-haven properties of bitcoin for those traditional assets. The main findings are as follows:(i)Bitcoin is positively linked to risk assets, including stocks, bonds and commodities, and negatively linked to the U.S. dollar, which is a typical safe-haven asset. Therefore, bitcoin is closer in nature to a risk asset than a safe-haven asset;(ii)The high short-term volatility and speculative nature of the bitcoin market makes its long-term correlation with other assets stronger than the short-term correlation;(iii)The positive linkage between bitcoin and risk assets increases sharply under extreme shocks (e.g., the outbreak of COVID-19 in early 2020);(iv)Bitcoin can hedge against the U.S. dollar, and in the long term, bitcoin can hedge against the Chinese stock market and act as a safe haven for the U.S. stock market and crude oil. However, for most other traditional assets, bitcoin is only an effective diversifier.

Our conclusions provide useful insights for market participants and policymakers. First, because bitcoin is closer in nature to a risk asset, investors should allocate to bitcoin as a risk diversifier for traditional risk assets such as stock, bond and commodity, rather than as a hedge, especially in times of extreme exogenous shocks. Second, the short-term high volatility and speculative nature of the bitcoin market leads to great uncertainty in the short-term price of bitcoin, while also undermining its short-term correlation with major financial assets, making bitcoin’s diversification, hedging and safe-haven properties vary across different time-frequency dimensions. This reminds bitcoin holders that it is important to distinguish between bitcoin holding periods. Investors who enter the bitcoin market should opt for long-term holdings as much as possible. Short-term speculation could expose them to significant investment risk and would likely result in large capital losses. Third, as uncertainties in global financial markets further increase in the post-epidemic era, policymakers and investors should keep paying attention to potential structural changes in the linkage between bitcoin and major asset prices under exogenous extreme shocks and the financial risks they may trigger. Finally, as the market for bitcoin trading is immature and the price is extremely unstable, individual investors should be discouraged from entering the cryptocurrency market represented by bitcoin, to protect the safety of their property.

## Figures and Tables

**Figure 1 entropy-24-01565-f001:**
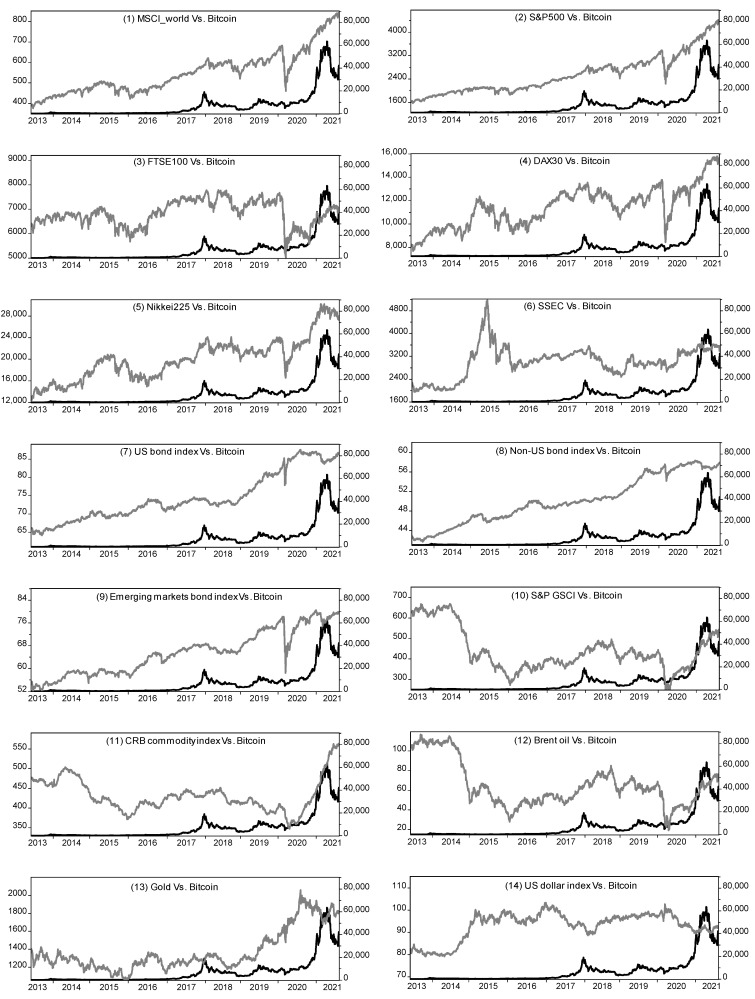
Time series of the prices of bitcoin versus various traditional financial assets. In each graph, the bitcoin price series is marked with a black line, corresponding to the right axis; the traditional asset price series is marked with a gray line, corresponding to the left axis.

**Figure 2 entropy-24-01565-f002:**
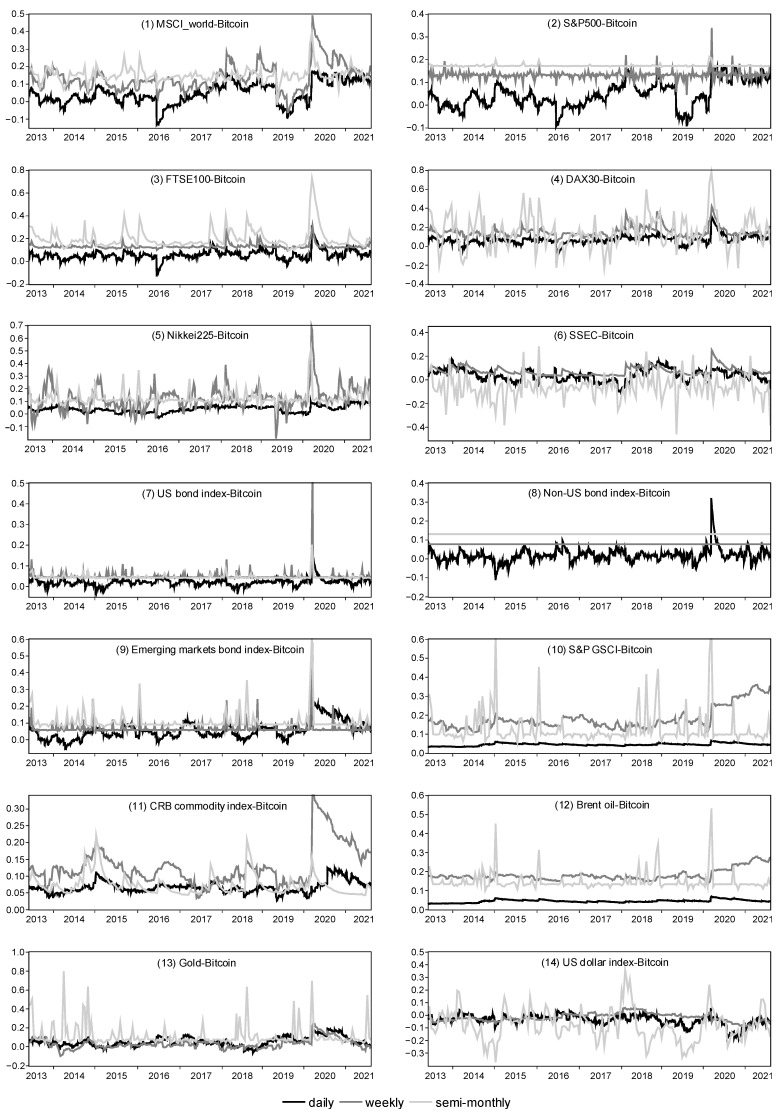
Daily, weekly and semi-monthly dynamic correlation coefficients between bitcoin and other assets.

**Figure 3 entropy-24-01565-f003:**
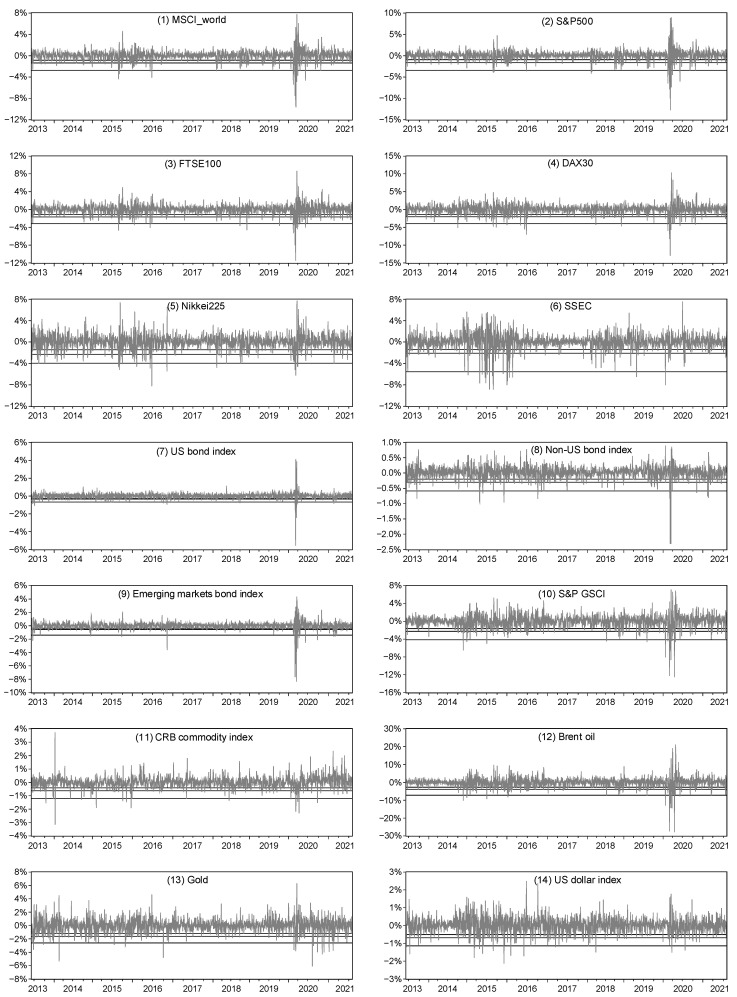
The 10%, 5% and 1% lower quartile values of 14 traditional asset returns (daily frequency). The three horizontal lines from top to bottom in each graph indicate the 10%, 5% and 1% lower quartile values, respectively.

**Table 1 entropy-24-01565-t001:** Descriptive statistics.

Variable	Mean	Median	Max.	Min.	Std. Dev.	Skewness	Kurtosis	J-B	*p* Value
Bitcoin	0.339	0.224	48.478	−49.397	5.685	−0.20	12.78	6795	0.0000
MSCI_world	0.045	0.083	7.793	−9.642	0.955	−1.44	23.52	30,451	0.0000
S&P500	0.059	0.084	8.968	−12.765	1.159	−1.23	24.35	32,771	0.0000
FTSE100	0.006	0.050	8.667	−11.512	1.114	−0.78	15.36	11,007	0.0000
DAX30	0.038	0.091	10.414	−13.055	1.350	−0.70	13.52	7989	0.0000
Nikkei225	0.045	0.060	7.731	−8.253	1.388	−0.12	6.75	1004	0.0000
SSEC	0.025	0.070	7.548	−8.873	1.499	−0.95	9.49	3249	0.0000
U.S. bond index	0.016	0.023	4.133	−5.592	0.316	−1.51	94.07	589,195	0.0000
Non-U.S. bond index	0.018	0.019	0.897	−2.320	0.213	−1.77	21.48	25,118	0.0000
Emerging markets bond index	0.021	0.049	4.345	−8.381	0.578	−3.93	63.28	262,235	0.0000
S&P GSCI	−0.009	0.061	7.115	−12.523	1.461	−0.93	11.82	5772	0.0000
CRB commodity index	0.010	0.000	3.726	−3.192	0.425	0.17	11.60	5255	0.0000
Brent oil	−0.020	0.083	21.115	−27.976	2.732	−0.85	21.61	24,769	0.0000
Gold	0.015	0.016	6.287	−6.120	1.044	−0.09	6.79	1020	0.0000
U.S. dollar index	0.006	0.003	2.495	−2.142	0.449	0.13	5.37	403	0.0000

Note: The J-B statistic is used to test the normality of the distribution of variables, and its null hypothesis is that the variable follows a normal distribution. The *p* value corresponds to the J-B test.

**Table 2 entropy-24-01565-t002:** Unit root test.

Variables	ADF			PP		
	Statistic	*p* Value	Result	Statistic	*p* Value	Result
Bitcoin	−21.28	0.0000	I(0)	−44.20	0.0001	I(0)
MSCI_world	−23.58	0.0000	I(0)	−41.42	0.0000	I(0)
S&P500	−27.57	0.0000	I(0)	−48.47	0.0001	I(0)
FTSE100	−41.24	0.0000	I(0)	−41.26	0.0000	I(0)
DAX30	−40.08	0.0000	I(0)	−40.07	0.0000	I(0)
Nikkei225	−42.12	0.0000	I(0)	−42.22	0.0000	I(0)
SSEC	−39.97	0.0000	I(0)	−39.98	0.0000	I(0)
U.S. bond index	−18.10	0.0000	I(0)	−42.74	0.0000	I(0)
Non-U.S. bond index	−41.59	0.0000	I(0)	−41.59	0.0000	I(0)
Emerging markets bond index	−14.82	0.0000	I(0)	−37.19	0.0000	I(0)
S&P GSCI	−41.67	0.0000	I(0)	−41.83	0.0000	I(0)
CRB commodity index	−20.19	0.0000	I(0)	−41.53	0.0000	I(0)
Brent oil	−40.07	0.0000	I(0)	−40.12	0.0000	I(0)
Gold	−40.85	0.0000	I(0)	−40.86	0.0000	I(0)
U.S. dollar index	−41.13	0.0000	I(0)	−41.13	0.0000	I(0)

**Table 3 entropy-24-01565-t003:** ARCH effect test.

Variables	ARCH(5)		ARCH(10)		ARCH(15)	
	Statistic	*p* Value	Statistic	*p* Value	Statistic	*p* Value
Bitcoin	89.13	0.0000	98.218	0.0000	105.45	0.0000
MSCI_world	580.03	0.0000	592.67	0.0000	628.76	0.0000
S&P500	714.77	0.0000	735.71	0.0000	807.19	0.0000
FTSE100	280.37	0.0000	409.82	0.0000	450.28	0.0000
DAX30	174.22	0.0000	298.34	0.0000	331.67	0.0000
Nikkei225	154.97	0.0000	213.87	0.0000	222.62	0.0000
SSEC	183.46	0.0000	193.38	0.0000	217.98	0.0000
U.S. bond index	377.26	0.0000	505.76	0.0000	536.61	0.0000
Non-U.S. bond index	445.02	0.0000	520.17	0.0000	532.95	0.0000
Emerging markets bond index	539.35	0.0000	705.25	0.0000	756.29	0.0000
S&P GSCI	140.46	0.0000	226.2	0.0000	241.21	0.0000
CRB commodity index	132.10	0.0000	134.20	0.0000	137.72	0.0000
Brent oil	117.79	0.0000	265.92	0.0000	311.46	0.0000
Gold	57.90	0.0000	98.66	0.0000	101.70	0.0000
U.S. dollar index	48.41	0.0000	60.45	0.0000	73.482	0.0000

Note: This table reports the results of the ARCH effect test at lags of order 5, 10 and 15. The null hypothesis for this test is “no ARCH effect”.

**Table 4 entropy-24-01565-t004:** Results of ADCC-GARCH estimation.

	Mean Equation		Variance Equation			ADCC Parameters			
	μ	φ	ω	α	β	a	b	g	υ
MSCI_world	0.0686 ***	0.1441 ***	0.0374 ***	0.1544 ***	0.7928 ***	0.0063	0.9849 ***	0.0000	4.0000 ***
	(0.0006)	(0.0000)	(0.0063)	(0.0001)	(0.0000)	(0.1662)	(0.0000)	(1.0000)	(0.0000)
Bitcoin	0.2708 ***	−0.0025	1.7513 **	0.1775 ***	0.7911 ***				
	(0.0078)	(0.9388)	(0.0157)	(0.0012)	(0.0000)				
S&P500	0.0996 ***	−0.0500 *	0.0733 ***	0.2741 ***	0.6756 ***	0.0066	0.9838 ***	0.0000	4.0000 ***
	(0.0000)	(0.0698)	(0.0004)	(0.0007)	(0.0000)	(0.8233)	(0.0000)	(1.0000)	(0.0000)
Bitcoin	0.2708 ***	−0.0025	1.7513 **	0.1775 ***	0.7911 ***				
	(0.0072)	(0.9385)	(0.0154)	(0.0011)	(0.0000)				
FTSE100	0.0212	0.0458	0.0507 **	0.1210 ***	0.8338 ***	0.0084	0.9552	0.0000	4.0000 ***
	(0.3310)	(0.1072)	(0.0197)	(0.0009)	(0.0000)	(0.8788)	(0.4951)	(1.0000)	(0.0000)
Bitcoin	0.2708 ***	−0.0025	1.7513 **	0.1775 ***	0.7911 ***				
	(0.0081)	(0.9390)	(0.0162)	(0.0011)	(0.0000)				
DAX30	0.0616 **	0.0208	0.0375 **	0.0783 ***	0.9000 ***	0.0078	0.9637 ***	0.0000	4.0000 ***
	(0.0243)	(0.4427)	(0.0164)	(0.0001)	(0.0000)	(0.2637)	(0.0000)	(1.0000)	(0.0000)
Bitcoin	0.2708 ***	−0.0025	1.7513 **	0.1775 ***	0.7911 ***				
	(0.0079)	(0.9388)	(0.0158)	(0.0012)	(0.0000)				
Nikkei225	0.0760 ***	−0.0367	0.0882 **	0.1146 ***	0.8396 ***	0.0030	0.9878 ***	0.0000	4.0000 ***
	(0.0069)	(0.1783)	(0.0318)	(0.0001)	(0.0000)	(0.3440)	(0.0000)	(1.0000)	(0.0000)
Bitcoin	0.2708 ***	−0.0025	1.7513 **	0.1775 ***	0.7911 ***				
	(0.0079)	(0.9388)	(0.0158)	(0.0012)	(0.0000)				
SSEC	0.0314	0.0266	0.0188	0.0778 ***	0.9177 ***	0.0069	0.9780 ***	0.0031	4.0000 ***
	(0.2363)	(0.3553)	(0.1164)	(0.0039)	(0.0000)	(0.1485)	(0.0000)	(0.5676)	(0.0000)
Bitcoin	0.2708 ***	−0.0025	1.7513 ***	0.1775 ***	0.7911 ***				
	(0.0081)	(0.9387)	(0.0158)	(0.0012)	(0.0000)				
U.S. bond index	0.0193 ***	−0.0025	0.0101 **	0.1479 **	0.6820 ***	0.0041	0.9383 ***	0.0000	4.1864 ***
	(0.0002)	(0.9266)	(0.0210)	(0.0117)	(0.0000)	(0.4438)	(0.0000)	(1.0000)	(0.0000)
Bitcoin	0.2708 ***	−0.0025	1.7513 **	0.1775 ***	0.7911 ***				
	(0.0080)	(0.9388)	(0.0158)	(0.0012)	(0.0000)				
Non-U.S. bond index	0.0238 ***	0.0388	0.0045 **	0.1619 **	0.7337 ***	0.0085	0.9324 ***	0.0000	4.0111 ***
	(0.0000)	(0.2261)	(0.0274)	(0.0102)	(0.0000)	(0.1982)	(0.0000)	(1.0000)	(0.0000)
Bitcoin	0.2708 ***	−0.0025	1.7513 **	0.1775 ***	0.7911 ***				
	(0.0080)	(0.9389)	(0.0158)	(0.0012)	(0.0000)				
Emerging markets bond index	0.0305 ***	0.0491	0.0103 ***	0.2980 ***	0.7010 ***	0.0058	0.9792 ***	0.0000	4.0000 ***
	(0.0003)	(0.1751)	(0.0025)	(0.0003)	(0.0000)	(0.3347)	(0.0000)	(1.0000)	(0.0000)
Bitcoin	0.2708 ***	−0.0025	1.7513 **	0.1775 ***	0.7911 ***				
	(0.0079)	(0.9387)	(0.0158)	(0.0012)	(0.0000)				
S&P GSCI	0.0174	0.0060	0.0333 **	0.0688 ***	0.9165 ***	0.0000	0.9927 ***	0.0009	4.0000 ***
	(0.5484)	(0.8154)	(0.0386)	(0.0000)	(0.0000)	(1.0000)	(0.0000)	(0.5566)	(0.0000)
Bitcoin	0.2708 ***	−0.0025	1.7513 **	0.1775 ***	0.7911 ***				
	(0.0079)	(0.9388)	(0.0158)	(0.0012)	(0.0000)				
CRB commodity index	0.0054	0.0847 ***	0.0103 *	0.0410 **	0.9015 ***	0.0024	0.9825 ***	0.0000	4.0000 ***
	(0.6288)	(0.0028)	(0.0999)	(0.0262)	(0.0000)	(0.5244)	(0.0000)	(1.0000)	(0.0000)
Bitcoin	0.2708 ***	−0.0025	1.7513 **	0.1775 ***	0.7911 ***				
	(0.0085)	(0.9387)	(0.0158)	(0.0012)	(0.0000)				
Brent oil	0.0478	−0.0190	0.0952 **	0.1205 ***	0.8761 ***	0.0000	0.9927	0.0010	4.0000 ***
	(0.3078)	(0.4767)	(0.0247)	(0.0000)	(0.0000)	(1.0000)	(0.5850)	(0.9954)	(0.0000)
Bitcoin	0.2708 **	−0.0025	1.7513 **	0.1775 ***	0.7911 ***				
	(0.0104)	(0.9376)	(0.0141)	(0.0013)	(0.0000)				
Gold	0.0087	0.0016	0.0076 **	0.0271 ***	0.9657 ***	0.0066	0.9780 ***	0.0000	4.0000 ***
	(0.7094)	(0.9521)	(0.0265)	(0.0000)	(0.0000)	(0.1151)	(0.0000)	(1.0000)	(0.0000)
Bitcoin	0.2708 ***	−0.0025	1.7513 **	0.1775 ***	0.7911 ***				
	(0.0080)	(0.9389)	(0.0157)	(0.0012)	(0.0000)				
U.S. dollar index	0.0047	−0.0060	0.0012 *	0.0355 ***	0.9589 ***	0.0076	0.9736 ***	0.0000	4.3210 ***
	(0.6250)	(0.8119)	(0.0778)	(0.0000)	(0.0000)	(0.1743)	(0.0000)	(1.0000)	(0.0000)
Bitcoin	0.2708 ***	−0.0025	1.7513 **	0.1775 ***	0.7911 ***				
	(0.0080)	(0.9389)	(0.0158)	(0.0012)	(0.0000)				

Note: This table reports the results of ADCC-GARCH estimation for the daily frequency sample. The *p* values are in parentheses. ***, ** and * indicate significance at the 1%, 5% and 10% levels, respectively.

**Table 5 entropy-24-01565-t005:** Regression results for the risk diversification, hedging and safe-haven properties of bitcoin for other assets (daily frequency).

	$10\%\;\mathrm{Quantile}\;(m_{1})$	$5\%\;\mathrm{Quantile}\;(m_{2})$	$1\%\;\mathrm{Quantile}\;(m_{3})$	$m_{0}$
MSCI_world	0.0049	0.0052	0.0141	0.0481 ***
	(0.4753)	(0.6080)	(0.3900)	(0.0000)
S&P500	0.0146 **	−0.0031	0.0247	0.0455 ***
	(0.0229)	(0.7416)	(0.1046)	(0.0000)
FTSE100	−0.0039	0.0067	0.0586 ***	0.0572 ***
	(0.4001)	(0.3302)	(0.0000)	(0.0000)
DAX30	0.0030	−0.0031	0.0406 ***	0.0679 ***
	(0.5659)	(0.6855)	(0.0010)	(0.0000)
Nikkei225	0.0007	−0.0025	0.0046	0.0390 ***
	(0.7960)	(0.5561)	(0.5064)	(0.0000)
SSEC	0.0076	−0.0056	−0.0176	0.0391 ***
	(0.1571)	(0.4772)	(0.1721)	(0.0000)
U.S. bond index	−0.0009	−0.0023	0.0240 ***	0.0212 ***
	(0.6568)	(0.4624)	(0.0000)	(0.0000)
Non-U.S. bond index	−0.0051	0.0031	0.0198 **	0.0184 ***
	(0.1968)	(0.5963)	(0.0363)	(0.0000)
Emerging markets bond index	0.0105 *	−0.0070	0.0916 ***	0.0532 ***
	(0.0632)	(0.3981)	(0.0000)	(0.0000)
S&P GSCI	0.0023 ***	0.0008	0.0043 **	0.0451 ***
	(0.0017)	(0.4658)	(0.0159)	(0.0000)
CRB commodity index	0.0003	−0.0027	−0.0025	0.0671 ***
	(0.8803)	(0.3298)	(0.5879)	(0.0000)
Brent oil	0.0022 ***	0.0032 ***	0.0039 **	0.0445 ***
	(0.0086)	(0.0073)	(0.0461)	(0.0000)
Gold	−0.0031	0.0091	0.0431 ***	0.0543 ***
	(0.5665)	(0.2536)	(0.0009)	(0.0000)
U.S. dollar index	−0.0016	0.0060	0.0072	−0.0373 ***
	(0.7287)	(0.3643)	(0.5032)	(0.0000)

Note: *p* values are in parentheses. ***, ** and * indicate significance at the 1%, 5% and 10% levels, respectively.

**Table 6 entropy-24-01565-t006:** Regression results for the risk diversification, hedging and safe-haven properties of bitcoin for other assets (weekly frequency).

	$10\%\;\mathrm{Quantile}\;(m_{1})$	$5\%\;\mathrm{Quantile}\;(m_{2})$	$1\%\;\mathrm{Quantile}\;(m_{3})$	$m_{0}$
MSCI_world	0.0263	−0.0138	0.0125	0.1357 ***
	(0.1283)	(0.5845)	(0.7494)	(0.0000)
S&P500	0.0009	0.0174 ***	−0.0206 **	0.1318 ***
	(0.8267)	(0.0037)	(0.0268)	(0.0000)
FTSE100	0.0025	−0.0021	0.0501 ***	0.1314 ***
	(0.6155)	(0.7775)	(0.0000)	(0.0000)
DAX30	−0.0005	0.0193	0.0519 *	0.1489 ***
	(0.9693)	(0.2701)	(0.0580)	(0.0000)
Nikkei225	−0.0215	−0.0080	0.0473	0.1290 ***
	(0.2985)	(0.7904)	(0.3139)	(0.0000)
SSEC	0.0070	−0.0112	−0.0050	0.0743 ***
	(0.3758)	(0.3320)	(0.7807)	(0.0000)
U.S. bond index	0.0010	−0.0031	0.0071	0.0491 ***
	(0.8678)	(0.7188)	(0.6011)	(0.0000)
Non-U.S. bond index	3.53 × 10^−9^	−9.88 × 10^−8^	4.40 × 10^−7^ ***	0.0772 ***
	(0.9610)	(0.3474)	(0.0075)	(0.0000)
Emerging markets bond index	0.0032	−0.0026	0.0755 ***	0.0665 ***
	(0.6917)	(0.8243)	(0.0001)	(0.0000)
S&P GSCI	−0.0109	0.0276	−0.0164	0.1805 ***
	(0.4083)	(0.1503)	(0.5828)	(0.0000)
CRB commodity index	0.0019	0.0024	0.0691 **	0.1274 ***
	(0.8851)	(0.9008)	(0.0195)	(0.0000)
Brent oil	−0.0124 *	0.0141	−0.0066	0.1828 ***
	(0.0571)	(0.1367)	(0.6563)	(0.0000)
Gold	0.0034	0.0162	0.0543 **	0.0303 ***
	(0.7694)	(0.3356)	(0.0395)	(0.0000)
U.S. dollar index	0.0058	−0.0041	−0.0084	−0.0182 ***
	(0.4148)	(0.6935)	(0.6027)	(0.0000)

Note: *p* values are in parentheses. ***, ** and * indicate significance at the 1%, 5% and 10% levels, respectively.

**Table 7 entropy-24-01565-t007:** Regression results for the risk diversification, hedging and safe-haven properties of bitcoin for other assets (semi-monthly frequency).

	$10\%\;\mathrm{Quantile}\;(m_{1})$	$5\%\;\mathrm{Quantile}\;(m_{2})$	$1\%\;\mathrm{Quantile}\;(m_{3})$	$m_{0}$
MSCI_world	0.0025	−0.0006	0.0942 ***	0.1505 ***
	(0.8494)	(0.9734)	(0.0032)	(0.0000)
S&P500	−0.0002	0.0100 ***	−0.0107 ***	0.1730 ***
	(0.8938)	(0.0000)	(0.0052)	(0.0000)
FTSE100	0.0215	−0.0177	0.1584 **	0.1990 ***
	(0.4422)	(0.6645)	(0.0210)	(0.0000)
DAX30	0.0922 *	−0.1048	0.3512 ***	0.1428 ***
	(0.0881)	(0.1836)	(0.0079)	(0.0000)
Nikkei225	0.0126	0.0064	0.1188 ***	0.1203 ***
	(0.4645)	(0.7992)	(0.0050)	(0.0000)
SSEC	0.0726 **	−0.0299	−0.0981	−0.0671 ***
	(0.0271)	(0.5299)	(0.2172)	(0.0000)
U.S. bond index	−0.0026	0.0098	−0.0110	0.0453 ***
	(0.5480)	(0.1217)	(0.2998)	(0.0000)
Non-U.S. bond index	−2.44 × 10^−9^	−5.92 × 10^−8^	−3.97 × 10^−8^	0.1307 ***
	(0.9245)	(0.1166)	(0.5259)	(0.0000)
Emerging markets bond index	−0.0095	0.0956 ***	0.0199	0.1025 ***
	(0.5822)	(0.0002)	(0.6362)	(0.0000)
S&P GSCI	0.0166	−0.0015	0.4531 ***	0.1210 ***
	(0.5002)	(0.9665)	(0.0000)	(0.0000)
CRB commodity index	0.0028	−0.0067	0.0659 ***	0.0736 ***
	(0.7768)	(0.6464)	(0.0074)	(0.0000)
Brent oil	0.0180	−0.0072	0.2724 ***	0.1429 ***
	(0.1671)	(0.7030)	(0.0000)	(0.0000)
Gold	−0.0393	−0.0011	0.1826 *	0.1183 ***
	(0.3131)	(0.9847)	(0.0551)	(0.0000)
U.S. dollar index	0.0506	−0.0428	0.0929	−0.0825 ***
	(0.2098)	(0.4673)	(0.3438)	(0.0000)

Note: *p* values are in parentheses. ***, ** and * indicate significance at the 1%, 5% and 10% levels, respectively.

## Data Availability

Data are available on request.

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
