# Peer review of "Dynamic Linkage between Bitcoin and Traditional Financial Assets: A Comparative Analysis of Different Time Frequencies"

_entropy, 2022, doi:10.3390/e24111565_

Round 1
Reviewer 1 Report
The paper examines the role of bitcoin as a safe haven asset for financial markets. Using DCC GARCH models, the authors find a positive linkages between bitcoin and both stocks, bonds, and commodity markets. A negative relationships is observed between bitcoin and USD index. Bitcoin is a diversifer asset for traditional financial assets and a hedge asset for US dollar index.
My comments are summarized as follows:
1) The contribution of the paper is weak. The authors should clearly demonstrate the contribution of this study compared to previous study.
https://doi.org/10.1016/j.najef.2022.101747
https://doi.org/10.1016/j.jmse.2019.09.001
https://doi.org/10.1016/j.irfa.2022.102121
https://doi.org/10.1016/j.resourpol.2020.101816
https://doi.org/10.1016/j.frl.2018.11.002
2) I suggest to create a separate section related to literature review and show your contribution versus previous empirical studies.
3) The empirical method is not well justified. Why not asymmetric DCC GARCH model?
4) Usually copula method is able to detect the safe haven asset during bearish market conditions.
5) Please compare your results with previous studies.
6) The authors should revised the legend of Figure 1.
Reviewer 2 Report
Title: Bitcoin: Risk Asset or Safe-haven Asset?
Comment 1. The paper does not clearly establish the criteria by which an asset is considered a risk asset or a safe-haven asset and such criteria are not analysed.
Comment 2. Further arguments are needed that the tools used in the paper are sufficient to answer the title question.
Comment 3. The contribution in the paper is limited to an empirical analysis using DCC-Garch and a regression function, where DCC is the dependent variable, tools developed in the scientific literature and appearing a long time ago. The work is much too didactic.
Comment 4. The paper does not present a literature review separated from the introduction and structured on the issue in the title. The bibliography is not relevant enough.
Comment 5. The methodology is very simplified and there are mathematical variables that are not defined.
Comment 6. For stationarity tests (ADF, PP) there are no interpretations of the t-statistic depending on the significance levels. It is not clear whether testing is done for level or first difference.
Comment 7. Mainly, an asset is considered a safe-haven if during the recession there is a preference (inclination) of investors towards such an asset (a necessary and not a sufficient condition). The choice of the analysis period is not argued in the paper based on this consideration.
Comment 8. Given the specifics of the journal, we should rather find the risk of the assets treated by means of entropy and not standard-deviation.
Comment 9. To examine risk diversification and volatility transmission more appropriate tools can be used, e.g: Wavelet Analysis.
Comment 10. The conclusions are not sufficiently structured, many of them are superfluous.
Round 2
Reviewer 1 Report
The authors have responded to the comments and suggestions.
Reviewer 2 Report
The paper is much too didactic.